# The Fast Lane of Hypoxic Adaptation: Glucose Transport Is Modulated via A HIF-Hydroxylase-AMPK-Axis in Jejunum Epithelium

**DOI:** 10.3390/ijms20204993

**Published:** 2019-10-09

**Authors:** Franziska Dengler, Gotthold Gäbel

**Affiliations:** Institute of Veterinary Physiology, University of Leipzig, 04103 Leipzig, Germany; gaebel@rz.uni-leipzig.de

**Keywords:** jejunum epithelium, hypoxia, AMPK, HIF hydroxylase, glucose transport, SGLT1, GLUT1, GLUT2

## Abstract

The intestinal epithelium is able to adapt to varying blood flow and, thus, oxygen availability. Still, the adaptation fails under pathologic situations. A better understanding of the mechanisms underlying the epithelial adaptation to hypoxia could help to improve the therapeutic approach. We hypothesized that the short-term adaptation to hypoxia is mediated via AMP-activated protein kinase (AMPK) and that it is coupled to the long-term adaptation by a common regulation mechanism, the HIF-hydroxylase enzymes. Further, we hypothesized the transepithelial transport of glucose to be part of this short-term adaptation. We conducted Ussing chamber studies using isolated lagomorph jejunum epithelium and cell culture experiments with CaCo-2 cells. The epithelia and cells were incubated under 100% and 21% O_2_, respectively, with the panhydroxylase inhibitor dimethyloxalylglycine (DMOG) or under 1% O_2_. We showed an activation of AMPK under hypoxia and after incubation with DMOG by Western blot. This could be related to functional effects like an impairment of Na^+^-coupled glucose transport. Inhibitor studies revealed a recruitment of glucose transporter 1 under hypoxia, but not after incubation with DMOG. Summing up, we showed an influence of hydroxylase enzymes on AMPK activity and similarities between hypoxia and the effects of hydroxylase inhibition on functional changes.

## 1. Introduction

Hypoxia is defined as “a state of cellular oxygen deprivation whereby the O_2_ requirements of a cell exceed that of available O_2_” [1]. The gastrointestinal tract is familiar with large oxygen gradients due to its localization at the border between the anaerobic lumen and the mesenterial vessels, as well as its varying perfusion through the latter. It has been established that the intestinal epithelium is able to adapt to these circumstances elaborately, existing in a state of ‘physiological hypoxia’ [1]. However, under pathologic conditions accompanied by hypoxia, such as strangulation of the small intestines, hypoperfusion, or inflammatory disease [2,3], this adaptation seems to fail. A better understanding of the adaptation reaction involved in epithelial survival could help to improve the therapeutic approach to hypoxia in a pathologic context.

The underlying adaptation mechanisms remain elusive. While hypoxia-inducible factor (HIF) is considered to be the ‘master regulator’ of adaptation to hypoxia [4], its main mode of action is on the transcriptional level and, thus, it takes some time for the adaptation reaction to be effective. However, for the epithelium’s survival under challenging circumstances such as a sudden onset of hypoxia, an adaptation within minutes is necessary. Recent research provided evidence for a fast adaptation to hypoxia taking place on the protein rather than on the gene expression level [5,6]. This quick reaction might at least bridge the critical time until HIF-mediated adaptation mechanisms are set to work and could, thus, be a critical point for early therapeutic intervention.

A crucial aspect of this adaptation seems to be the supply of the enterocytes with energy sources, i.e., mainly glucose and amino acids. Kles and Tappenden [7] reported that the transepithelial transport of glucose was impaired under hypoxia, while glutamine transport was not. In accordance with that, it has been shown that the activity of the sodium-coupled glucose transporter SGLT1 was diminished under hypoxia in jejunum epithelium [5,8,9]. This downregulation appears to be a targeted and reasoned adaptation rather than a sign of cellular failure. Accordingly, the total transepithelial transport of glucose is sustained in spite of the missing import via SGLT1 [5]. Some authors postulate a recruitment of glucose transporter 2 (GLUT2) to the apical membrane in order to enhance the energy-independent uptake of glucose [9,10,11]. This switch from energy-dependent uptake to facilitated diffusion infers the advantage that the glucose supply could be sustained while less ATP is consumed by the enterocytes for its uptake. Thus, the available ATP originating rather from anaerobic glycolysis under hypoxia, which is way less effective compared to aerobic glycolysis, is used economically.

While the exact transport systems involved in this adaptation are still under discussion and need further investigation, it is also not clear yet by which intracellular mechanism this fast adaptation is mediated. Our own previous work indicated an involvement of the AMP-activated protein kinase (AMPK) in the modulation of transepithelial glucose transport in the lagomorph jejunum epithelium [5]. An activation of AMPK under hypoxia has also been postulated in human airway epithelial cells, HeLa cells, as well as prostate carcinoma cells [12,13,14]. AMPK is an ubiquitously expressed serine/threonine kinase monitoring cellular energy levels [15,16]. It is activated by phosphorylation on Thr172 of the α subunit [16]. Binding of AMP to the γ subunit promotes this phosphorylation of the α subunit that can be mediated by at least two different upstream kinases: liver kinase B (LKB) 1 and Ca^2+^/calmodulin-dependent protein kinase kinase (CAMKK) 2 [17,18]. Additionally, the allosteric binding of AMP prevents it from dephosphorylation, thus coupling its activity tightly to cellular energy levels [19]. In general, AMPK deactivates anabolic, energy-consuming processes and activates catabolic, energy-delivering processes, e.g., by phosphorylation of acetyl-CoA carboxylase (ACC), the rate limiting enzyme for fatty acid synthesis, and of phosphofructokinase, enhancing glycolysis, respectively, thus ameliorating the cellular ATP supply [17,20].

However, our own and others’ results strongly indicate that hypoxia does not lead to an immediate and complete depletion of ATP, although aerobic glycolysis might be disabled under hypoxia [5,21]. Thus, the activation of AMPK under hypoxia is not necessarily initiated by an increased AMP/ATP ratio, but alternate mechanisms activating AMPK under hypoxia have to be taken into account in the intestinal epithelial cells. Thus, there might be energy-independent ways of activating AMPK that have not been elucidated yet, anticipating the critical situation before the cells are damaged severely.

One of them could be the HIF-prolyl-4-hydroxylases (PHDs). PHDs are the central sensors of cellular oxygen supply and control the adaptation to hypoxia by mediating an oxygen-dependent hydroxylation of the cytoplasmic α subunit of HIF [22]. This oxygen-induced hydroxylation leads to the degradation of HIFα. Under hypoxia, the hydroxylation reaction ceases due to the lack of the cofactor oxygen and, thus, HIFα is stabilized and able to translocate into the nucleus where it dimerizes with the β subunit and modulates the transcription of >300 genes involved in metabolism, angiogenesis, and proliferation, in order to adjust the cellular functions to the hypoxic conditions [23,24]. Additionally, a second oxygen-dependent repression of HIF transcriptional activity is regulated by asparaginyl hydroxylation: another hydroxylase (factor inhibiting HIF, FIH) hydroxylates an asparagine residue within HIFα subunits, resulting in steric inhibition of its interaction with the transcriptional co-activator p300/CBP, thereby inhibiting HIF-dependent transcription [25,26].

Besides regulating HIF, there are already studies hinting at an activation of AMPK by FIH inhibition in brown adipose tissue [27]. A coupling of the long-term (HIF-mediated) and the short-term (AMPK-mediated) adaptation by a common pathway would be the ideal way of coordinating the adaptation reactions. As outlined above, HIF-hydroxylases might be that common pathway. Thus, we hypothesize AMPK under the control of PHDs/FIH to be the “fast lane” in hypoxic adaptation of the intestinal epithelium.

Therefore, we wanted to investigate if hydroxylase enzymes play a role in the activation of AMPK under hypoxia in jejunum epithelium and how functional effects regarding the alteration of glucose transport are implemented. Therefore, we employed a combined approach using the advantages of the functional Ussing chamber technique and a cell line enabling a more detailed molecular examination. In this study, we show proof of an activation of AMPK by the panhydroxylase inhibitor dimethyloxalylglycine (DMOG) and similar effects of hydroxylase inhibition and hypoxia on glucose transport across jejunum epithelium.

## 2. Results

### 2.1. The Panhydroxylase Inhibitor DMOG Mimics the Effect of Hypoxia on SGLT1 Activity

Previously, we showed a reduced activity of SGLT1 under hypoxia in lagomorph jejunum epithelium and an involvement of AMPK in its downregulation [5]. In order to test our hypothesis that AMPK is under the control of the PHDs and thus responsive to hypoxia, we preincubated the epithelia with the panhydroxylase inhibitor DMOG before submitting part of the epithelia to hypoxia. Subsequently, we added its inhibitor, phlorizin [28], to the mucosal buffer solution to determine the activity of SGLT1. Addition of phlorizin led to a drop in transepithelial short-circuit current (I_sc_). To correct the decrease in I_sc_ induced by phlorizin, i.e., the SGLT1-dependent current, for the time-dependent decrease, we determined Δm, i.e., the difference between the slopes of the I_sc_ course before and after mucosal addition of phlorizin (see Figure 1a), and compared the effects of O_2_-supply and DMOG preincubation using a one-way repeated measurements ANOVA (see Figure 1b).

We found a significant decrease of Δm under hypoxia compared to control conditions (*p* < 0.05) and after preincubation with DMOG compared to the respective group of similarly gassed epithelia incubated without DMOG (*p* < 0.05). However, there was no difference between epithelia incubated under hypoxia only and with DMOG only (under 100% O_2_ gassing). This indicates that inhibition of the PHDs mimics the effect of hypoxia on the activity of SGLT1. The larger decrease of Δm by DMOG incubation under hypoxia (39 ± 25 µEq cm^−2^ h^−1^ min^−1^) compared to hypoxia alone (190 ± 172 µEq cm^−2^ h^−1^ min^−1^) hints at a stronger or additive effect of DMOG compared to hypoxia.

### 2.2. Transepithelial Glucose Transport Is Sensitive to STF-31 under Hypoxia Only

With the activity of SGLT1 being decreased under hypoxia, we were wondering if other transporters may sustain the transepithelial transport of glucose, as we could show before [5]. To investigate this, we measured transepithelial fluxes of ^14^C-glucose and incubated the epithelia with inhibitors for either SGLT1 (phlorizin), GLUT1 (STF-31), or GLUT2 (cytochalasin B) on the mucosal side. Comparison of the groups revealed (i) no difference between J_ms_^glucose^ under hypoxia and control conditions, i.e., 100% O_2_, and (ii) a sensitivity of transepithelial glucose transport to phlorizin and cytochalasin B under both control and hypoxic conditions (Figure 2). Under hypoxia, however, we also observed a decreased flux rate after preincubation with the GLUT1 inhibitor STF-31, which was not observed under control conditions (Figure 2). This indicates a more important role for GLUT1 in transepithelial glucose transport under hypoxia.

Based on this observation, we intended to test whether preincubating the epithelia with DMOG would mimic the effects of hypoxia in this regard as well. However, there was no effect of STF-31 in these experiments (Figure 3). It must be taken into consideration though, that J_ms_^glucose^ was lower in the epithelia incubated with DMOG (Figure 3) compared to the experiments without DMOG preincubation before (Figure 2), i.e., on the same level J_ms_^glucose^ reached after addition of the inhibitors in the previous experimental series. Thus, there might be other effects of DMOG on transepithelial glucose transport besides a possible activation of AMPK, maybe masking the effect of STF-31.

### 2.3. Panhydroxylase Inhibiton Results in an Activation of AMPK

To investigate the effects of panhydroxylase inhibition further on the cellular level, we used a CaCo-2 cell line, which is known to resemble the small intestinal epithelium [29]. The cells were incubated with or without 3 mM DMOG (under 21% O_2_) or in 1% O_2_ for 0.5, 1, 3, 6, and 24 h. Then, the activation of AMPK was assessed by measuring the phosphorylation grade of AMPKα as well as its target ACC.

The ratio of pACC/ACC measured by Western blot was increased significantly after 0.5 h of incubation with both DMOG and under hypoxia. It was also increased after 1 h of incubation with DMOG, but not after 1 h of hypoxia (*p* < 0.01, one-way repeated measurements ANOVA). Longer incubation times did not increase the pACC/ACC ratio compared to the control group incubated at 21% O_2_ (see Figure 4a). There was no difference after incubation for 3, 6, or 24 h. A similar time course was observed for the pAMPKα/AMPKα ratio. After 0.5 and 1 h of DMOG treatment, the pAMPKα/AMPKα ratio was significantly increased compared to the control group incubated at 21% O_2_ but also compared to the hypoxic group (Figure 4b, *p* < 0.05, one-way repeated measurements ANOVA). We can only speculate why the activation indicated by the increased pACC/ACC ratio after hypoxic incubation is not mirrored in the pAMPKα/AMPKα ratio as well. Assuming that the phosphorylation of AMPK precedes the phosphorylation of its targets, the phosphorylation of AMPKα may have been only transient and already gone after 0.5 h of incubation under hypoxia, but with DMOG being a more potent activator it may still have been visible under this treatment.

### 2.4. Total Protein Expression of SGLT1, GLUT1, and GLUT2 Is Not Affected by DMOG or Hypoxia but Phosphorylation of GLUT1 Is Increased by Hypoxia Only

Looking for an explanation for the previously observed decreased SGLT1 activity under hypoxia, we also assessed the protein expression of this transporter in CaCo-2 cells, along with that of GLUT1 and 2 by Western blot. In the total protein extracted from the cells (treatment of 0.5 and 1 h, see above), we did not detect any differences in the protein expression of these transporters (Figure 5).

However, when using an antibody specific for GLUT1 phosphorylated at Ser226 (pGLUT1), we could identify a significant increase after 0.5 h of hypoxia compared to cells incubated at 21% O_2_ (Figure 6, paired t-test, *p* < 0.05). DMOG, in contrast, had no effect.

### 2.5. mRNA Expression of SGLT1, GLUT1, and GLUT2 Is Not Affected by DMOG or Hypoxia

Besides protein expression, we also assessed the mRNA expression of SGLT1, GLUT1 and 2 in CaCo-2 cells incubated under 21% O_2_ with or without DMOG or under hypoxia, as well as in isolated lagomorph jejunum epithelium after incubation in Ussing chambers under 100% O_2_ or 1% O_2_ with or without DMOG. We observed no significant changes in the gene expression after 1 h hypoxia and incubation with DMOG, respectively (Figure 7).

## 3. Discussion

In this study, we aimed to investigate short-term adaptive reactions and their underlying regulative mechanism under hypoxia in the intestinal epithelium. We hypothesized that a quick adaptation reaction mediated on the protein level via AMPK precedes a more sustainable HIF-mediated adaptation on the transcriptional level in order to secure crucial functions for the enterocytes’ survival. Furthermore, we postulated the import of glucose to be one of these crucial functions and, thus, a modulation of the transepithelial transport mechanisms for glucose by AMPK under hypoxia.

We reproduced previous results that showed a reduced activity of SGLT1 in small intestinal epithelium under hypoxia [5,7]. Additionally, we showed that preincubation with DMOG has a similar effect on the SGLT1-dependent I_sc_ (Figure 1), supporting our mechanistic hypothesis that AMPK, which has been shown to downregulate SGLT1 under hypoxia before [5], is under the control of hydroxylases.

Still, it is not clear yet, how the transepithelial transport of glucose is mediated under hypoxia. In spite of a reduced SGLT1 activity, the transepithelial transport of glucose has been reported to be stable under hypoxia [5,30], which was confirmed by the experiments presented herein (Figure 2). Looking for the transport mechanism responsible for sustaining the transepithelial transport of glucose, we used several inhibitors for flux measurements of ^14^C-glucose (Figure 2). We found inhibitory effects of the SGLT1-inhibitor phlorizin and the GLUT2-inhibitor cytochalasin B to be similar both under 1% and 100% O_2_. With respect to the aforementioned reduced activity of SGLT1 under hypoxia and the established view of GLUT2 being localized at the basolateral membrane only, this was quite surprising and cannot be explained sufficiently here. One might well doubt the specificity of these inhibitors, especially with cytochalasin B acting as a disruptor of the cytoskeleton [31,32]. Casting this aside, however, the most important finding of this experiment was an effect of STF-31, a GLUT1-inhibitor [33,34], under hypoxia only. This indicates that SGLT1 is indeed supplemented by another transport mechanism, i.e., GLUT1, although not by GLUT2 as postulated before [10,30,35]. This fits very well with GLUT1 gene transcription also being a HIF target [23], thus the long-term adaptation would support the modulations initiated in the short run. As outlined above, the replacement of glucose transport via SGLT1 by GLUT1 would be energetically less expensive for the cells. Additionally, it may be assumed that under hypoxia the enterocytes consume more glucose themselves in order to generate ATP by anaerobic glycolysis and, thus, the concentration gradient necessary for the import of glucose via GLUT1 is kept up. Although we found the transepithelial transport of ^14^C-glucose to be stable under hypoxia and control conditions, it has to be kept in mind that the measurement of ^14^C cannot discriminate between the original substrate glucose and its metabolites that may be extruded to the serosal side.

To explain the modifications observed in transepithelial glucose transport, we assessed the gene and protein expression of SGLT1, GLUT1 and 2. However, we did not observe changes, neither on the protein (Figure 5) nor the gene expression level (Figure 7) of any of the transporters. The lack of differences in the mRNA expression after 1 h of incubation is not surprising, as HIF-dependent gene transcription regulation becomes effective one hour after the onset of hypoxia at the earliest. Nonetheless this finding is in accordance with others’ observations showing no changes in total protein expression of SGLT1 after hypoxia [7] or under activation of the AMPK pathway [36]. However, differences in SGLT1 brush border membrane localization in rat jejunum epithelium after simulation of hypoxia in vivo [7], as well as an increased amount of GLUT2 in the brush border membrane after incubation with an AMPK agonist [36] have been described. This is in accordance with our assumption that AMPK mediates these functional changes, which is known to be effective via phosphorylation of its targets. Therefore, the total protein concentration might remain unchanged while the amount of phosphorylated, and thus membrane-bound, transporter is modulated. A modulation of existing transporter proteins is reflected by our findings of an increased phosphorylation of GLUT1 after 0.5 h of hypoxia (Figure 6). Lee et al. (2015) previously described a phosphorylation-dependent enhancement of GLUT1-mediated glucose transport in erythrocytes and endothelial cells [37]. The observation that pGLUT1 is increased only after hypoxic incubation but not by DMOG is in accordance with our results from the functional studies, which also indicated a recruitment of GLUT1 only after hypoxia but not after DMOG pretreatment. Thus, AMPK is either activated by other means than PHDs under hypoxia or another, hydroxylase-independent, pathway is responsible for the recruitment of GLUT1 under hypoxia. A phosphorylation of SGLT1 should be investigated further in future studies, although it might be limited by the availability of commercial antibodies.

Apart from the recruitment of GLUT1, we also found indicators for an activation of AMPK under hypoxia controlled by PHDs/FIH. We used the panhydroxylase inhibitor, DMOG, in functional studies, as well as studies on the molecular level. Several results favor our hypothesis: We could achieve an activation of AMPK by incubation with DMOG in CaCo-2 cells (Figure 4), indicating that AMPK activity is actually controlled by hydroxylation under normoxia and could thus be regulated along with HIF. Similar results have been obtained before in neonatal rat cardiomyocytes incubated with DMOG, in which besides an activation of AMPK, a stabilization of HIF could also be shown [38]. In contrast to our study, however, Yan et al. (2012) found the activation of AMPK and HIF not only for a short period of incubation with DMOG but for up to 9 h [38]. We could detect only a transient activation of AMPK under hypoxia/DMOG incubation lasting for 1 h. Other studies also found a maximum of AMPK activity after 60 min of hypoxia followed by a subsequent decline [13,39], showing a coherence of the vanishing AMPK activation with a gradually increasing HIF stabilization [13]. All these observations support our hypothesis of AMPK being the quick way for adaptation until HIF takes over. However, it has to be kept in mind that DMOG is a rather unspecific inhibitor and further studies on the PHD-AMPK axis with more specific inhibitors or knock outs for the PHDs are needed to prove this.

While Tan et al. [14] could only confirm the “classical” way for AMPK activation, i.e., an increased AMP:ATP ratio, under hypoxia, others postulate additional modes. CAMKK is considered to be activated via reactive oxygen species generated under hypoxia [13,40,41] or—supporting our hypothesis—via hydroxylation by the PHDs in cardiomyocytes and epithelial cell lines [38]. An involvement of CAMKK is further strengthened by the finding that silencing PHD2 in cardiomyocytes induced an activation of AMPK similar to the effect of DMOG, but had no effect when a Ca^2+^-chelator was added [38]. In brown adipose tissue, in contrast, FIH was shown to regulate AMPK by direct hydroxylation [27]. While not much is known regarding the proline-hydroxylation of non-HIF targets so far, asparagine hydroxylation by FIH is more common and has been demonstrated for multiple non-HIF proteins, including pathways involving AMPK [26], and might thus be more likely.

Supporting our view, we found similar effects of DMOG and hypoxia on the gene and protein expression of the glucose transporters (Figure 5 and Figure 7) and on the activity of SGLT1 measured in isolated lagomorph jejunum epithelia in Ussing chambers (Figure 1).

When investigating the functional effects of DMOG on SGLT1 activity, however, we found a more pronounced effect of DMOG than that of hypoxia alone (Figure 1). This could be due to a higher impact of the inhibitor compared to the 1% oxygen gassing, but also an additive effect of both might be taken into consideration. A higher impact of DMOG is also supported by the observation that the activation of AMPK is slightly more pronounced and lasts longer than under hypoxia in the CaCo-2 cells (Figure 4). Yan et al. (2012) reported a smaller effect of knocking out PHD2 compared to an application of the panhydroxylase inhibitor on AMPK activity [38]. This indicates that there might be more than one hydroxylase enzyme working redundantly. However, we used a rather unspecific panhydroxylase inhibitor and further studies employing more specific inhibitors, distinct knock outs, or siRNA approaches are necessary to identify specific enzymes involved in the regulation of AMPK, as well as its crosstalk with HIF.

Summing up, we could show an influence of hydroxylase enzymes on AMPK activity and similarities between hypoxia and the effects of hydroxylase inhibition. However, the effects differ in some aspects, again documenting that hypoxic signaling is no simple story but a complex interplay of numerous factors that might even be interchangeable. The inhibition of PHDs is considered an interesting target for the treatment of several pathological conditions, like anemia originating from chronic kidney disease or inflammatory bowel disease [22]. Thus, a profound knowledge of other pathways controlled by PHDs that might also be targeted by clinical drugs aimed at the regulation of HIF, is necessary to understand their potential side effects, both positive and negative.

## 4. Materials and Methods

### 4.1. Ussing Chamber Experiments

#### 4.1.1. Animals and Tissue Sampling

The Ussing chamber experiments were conducted as described before [5]. Briefly, 6- to 24-month-old rabbits (Vienna Blue and White or New Zealand Red Rabbit) of both sexes were fed a standard diet consisting of water and hay ad libitum, carrots, and a standard rabbit concentrate formula for at least ten days before slaughter. The animals were killed by exsanguination after captive bolt stunning. Then, the abdominal cavity was opened, and the mid-jejunum was excised, opened along the mesenteric border, and rinsed and submerged in 37 °C warm oxygenated buffer solution. After transport to the laboratory, the epithelium was stripped off the muscle layers manually and mounted in Ussing chambers as described previously [5]. The area exposed accounted for 1.1 cm^2^. The epithelia were allowed to equilibrate in the system for at least 30 min before conducting the experiment.

The experiments were conducted in accordance with the EU directive 2010/63/EU and the German legislation on the protection of animals. They were reported to the Landesdirektion Leipzig as T 46/14 (date of approval 21 October 2014).

#### 4.1.2. Buffer Solutions and Gassing

The buffer solutions were prepared with chemicals obtained from Sigma-Aldrich (Darmstadt, Germany), Carl Roth (Karlsruhe, Germany), VWR (Darmstadt, Germany), or Merck (Darmstadt, Germany), unless stated otherwise. The gasses were procured from Air Liquide (Düsseldorf, Germany).

In all of the experiments, a basal buffer solution consisting of 120 mM NaCl, 5.5 mM KCl, 1.25 mM CaCl_2_, 1.25 mM MgCl_2_, 0.6 mM NaH_2_PO_4_, 2.4 mM Na_2_HPO_4_, 10 mM glucose, 5 mM l-glutamine, and 10 mM HEPES was used for rinsing, preparation, and transport of the epithelia. All other buffer solutions were based on this basal buffer solution, except glucose, which was reduced to 3 mM for the flux buffer solution or replaced by 3 mM alpha-methyl-glucose (AMG) for the electrophysiological measurements. Mannitol was used to adjust the osmolarity to 280 ± 5 mOsm/L. The pH was adjusted to 7.4 with 1N HCl or NaOH. All buffer solutions were gassed with 100% oxygen. Hypoxia was simulated by changing the gassing from 100% O_2_ to 99% N_2_ plus 1% O_2_ after the equilibration period.

The use of 100% oxygen for gassing the control group might be considered as hyper- rather than normoxic. However, it has to be kept in mind, that oxygen has to be dissolved physically in the buffer solution and cannot be transported as effectively as in vivo. Therefore, it is assumed that the actual amount of oxygen near and inside the tissue is less than 100%. The diffusion distance is further increased by additional layers besides the ‘isolated’ epithelium (basal membrane, mucus, connective tissue) in contrast to cultured cells growing in monolayer. Therefore, only a small part of the 100% oxygen used for gassing will reach the epithelial cells. Moreover, the functional differences observed in our experiments indicate that there is a biologically significant difference in oxygen supply.

#### 4.1.3. Electrical Measurements

Electrical measurements were taken continuously with the aid of a computer-controlled voltage clamp device (Ingenieurbüro für Mess- und Datentechnik, Dipl.-Ing. K. Mußler, Aachen, Germany). All experiments were conducted under short-circuit conditions. The short-circuit current (I_sc_) and transepithelial tissue conductance (G_t_) were calculated computationally as described before [5].

In each experiment, the different treatments were assigned to the individual epithelia within one animal according to their G_t_ so that at the end of an experimental series, the mean value of G_t_ was similar in all treatment groups.

For the measurement of SGLT1 activity, epithelia were preincubated with 2 mM DMOG for 15 min before subjecting part of the epithelia to hypoxia. After 45 min of hypoxia, 0.2 mM phlorizin, an inhibitor of SGLT1 [28], was added to the mucosal buffer solution and the difference in the slope of I_sc_ before and after this addition (i.e., the loss of net charge transport across the epithelium) was measured.

The slope of I_sc_, i.e., the decrease in I_sc_ over time, was calculated using the formula m = I_sc_1 − I_sc_2/t1 − t2 for one minute before (m_b_) and one minute after (m_a_) the addition of phlorizin [5]. The difference Δm = m_b_ − m_a_ (µEq cm^−2^ h^−1^ min^−1^) was calculated and used for statistical comparison. By comparing Δm, the drop in I_sc_ could be assessed independently from the individual epithelium’s absolute I_sc_ value, and thus, the effect of SGLT1 inhibition could be evaluated (see Figure 1a).

After dismounting the epithelia, they were washed in PBS, snap-frozen in liquid nitrogen, and stored at −80 °C for RNA extraction and assessment of the gene expression (see below).

#### 4.1.4. Flux Rate Measurements

The transepithelial transport (‘flux rates’) of radioactively labelled ^14^C-glucose was measured from the mucosal (‘hot side’, 4 kBq/mL) to the serosal (‘cold side’) side of the epithelium (J_ms_^glucose^).

The buffer solutions for these experiments contained 3 mM glucose on both sides of the epithelium. To trace the transport of the substance across the epithelium, the radioactively labelled substrate was added to the mucosal side, and samples (0.8 mL) were taken from the serosal side every 30 min. The volume removed was replaced with the respective buffer solution and corrected for in flux ratio calculations. The amount of transported substrate could be derived from these measurements. A first flux period was measured without inhibitors to ensure similar flux rates across all epithelia. Then, part of the epithelia was submitted to hypoxia and one of the inhibitors was added to the mucosal side and preincubated for 30 min. In order to identify the transporters involved in the transport of glucose across the epithelium, we applied 0.2 mM phlorizin, 0.5 mM STF-31, and 0.2 mM cytochalasin B as inhibitors for SGLT1, GLUT1, and GLUT2, respectively. The inhibitors were prepared as stock solutions, so that only small volumes were added to the buffer solutions. Phlorizin was dissolved in ethanol and cytochalasin B and STF-31 in DMSO. The solvents were also added to the control groups, which were pooled to one group because there was no difference in J_ms_^glucose^.

In a second experiment, we also tested the effect of preincubating the epithelia with the panhydroxylase inhibitor DMOG (2 mM, dissolved in distilled water) on the STF-31-sensitive J_ms_^glucose^. Here, DMOG was added after the initial equilibration period of approximately 30 min, otherwise the same protocol as described above was used.

The activity of radioisotopes was measured as decays per minute (dpm) by photometric measurements (Liquid Scintillation Analyzer Tri-Carb 2810, Perkin Elmer Inc., Waltham, MA, USA). The corresponding amount of substrate was calculated using a simple ratio equation. In each experiment, two samples (0.1 mL) were taken from the ‘hot’ side, to which the radioactively labelled substrate was added, and their averaged dpm values were used to calculate the specific activity used. Aquasafe^®^ 300 Plus (Fa. Zinsser Analytic, Eschborn, Germany) was used as scintillation fluid. ^14^C-glucose was obtained from Hartmann Analytic GmbH (Braunschweig, Germany).

### 4.2. Cell Culture

CaCo-2 cells were used for cell culture experiments to gain further insights into the mechanisms underlying the functional changes observed in the Ussing chamber experiments. The cells were a generous gift from H.Y. Naim, Institute for Biochemistry, University of Veterinary Medicine, Hannover, Foundation, Germany. They were incubated in a humidified 5% CO_2_ air atmosphere at 37 °C in Dulbecco’s modified Eagle’s medium (DMEM) with high glucose content supplemented with 10% fetal calf serum (FCS), 100 U/mL penicillin/streptomycin and 3 mM L-glutamine. Passages 20–30 were used for the experiments. Cells were seeded either in cell culture flasks of 25 cm^2^ for protein extraction or on 12-well plates (both greiner bio-one, Frickenhausen, Germany) for RT-qPCR.

All chemicals for cell culture were obtained from Sigma-Aldrich (Darmstatdt, Germany).

When the cells were near confluency (>95%), they were washed with PBS once, fresh medium was added, and the respective treatment was applied. DMOG was dissolved in DMEM and added to the fresh medium from a stock solution to an end concentration of 3 mM. The medium for the epithelia to be incubated under 1% O_2_ was preequilibrated for 20 min in the hypoxic incubator and the cells were put in the incubator without lid in order to support a free exchange with the incubator’s atmosphere.

After the incubation time of 0.5, 1, 3, 6, or 24 h, the cells were washed with ice-cold PBS once and detached mechanically using a cell scraper. All steps were conducted on ice.

For total protein extraction, the cell suspension (in PBS) was centrifuged at 800× *g* and 4 °C for 5 min and the pellet was resuspended in 500 µL of a lysis buffer consisting of 10 mM EDTA, 4 mM EGTA, 50 mM TRIS buffer, 100 mM β-glycerin-phosphate-disodium pentahydrate, 0.1% Triton X-100, 15 mM sodium orthovanadate, 100 mM sodium pyrophosphate tetrabasic decahydrate, and 2.5 mM NaF (pH 7.4) with a protease and phosphatase inhibitor (100X Halt™ protease and phosphatase inhibitor cocktail, Thermo Fisher Scientific, Dreieich, Germany). The protein concentration was measured with a Tecan Spectra Rainbow Microplate Reader (Tecan Deutschland, Crailsheim, Germany) using the bicinchoninic acid method and bovine serum albumin as standard.

For the evaluation of gene expression, the cells were also spun down at 800× *g* for 5 min and the pellet was stored at −80 °C until further processing.

### 4.3. Western Blot Analysis

For Western blot analysis of AMPKα, ACC, and their phosphorylated forms (pAMPKα, pACC), as well as the protein expression of the glucose transporters, the protein samples were separated by sodium dodecyl sulphate-polyacrylamide gel electrophoresis (SDS-PAGE) using 10 µg protein/well. Subsequently, the samples were transferred onto a nitrocellulose membrane (Carl Roth, Karlsruhe, Germany) using the Mini-Protean^®^ system (Bio-rad laboratories, Feldkirchen, Germany). The membrane was preincubated in 5% bovine serum albumin in TRIS-buffered saline containing 0.2% Tween-20 (TBST) at 4 °C over night. On the next day, it was incubated with the primary antibodies (see Table 1) at room temperature with gentle shaking for two hours. After washing with TBST five times, the membranes were incubated with an HRP-coupled secondary antibody (see Table 1) at room temperature with gentle shaking for one hour. Subsequently, the membranes were rinsed again with TBST five times and once with TBS, then the signal was detected by enhanced chemiluminescence using a G:BOX Chemi XT4 and analyzed with the GeneTools© software (both Syngene, Cambridge, UK). β-Actin or Villin was used as loading control for normalization of each blot.

### 4.4. Two-Step Real-Time Reverse Transcriptase Polymerase Chain Reaction (RT-qPCR)

The RNA was extracted from 10 mg of tissue samples and frozen cell pellets, respectively, using the ReliaPrep™ RNA Miniprep System (Promega, Mannheim, Germany) following the manufacturer’s protocol. The RNA concentration and quality were determined with the aid of a spectrophotometer (BioPhotometer, Eppendorf, Wesseling-Berzdorf, Germany). In total, 1 µg of high-quality RNA was used for cDNA synthesis using the GoScriptTM Reverse Transcriptase Kit (Promega, Mannheim, Germany) according to the manufacturer’s instructions using a MJ Research PTC-200 Peltier Thermal Cycler (Bio-Rad, Feldkirchen, Germany).

For qPCR, the resulting cDNA was diluted 1:20 and 2 µL were used in a 20 µL reaction volume containing 10 µL of a ready-to-use premix of SYBR Green I dye, dNTPs, stabilizers and enhancers (GoTaq^®^, Promega, Mannheim, Germany), 112 nM primer mix, and DNase-free water. These mixtures were pipetted in strip tubes (0.1 mL Strips, LTF Labortechnik, Wasserburg, Germany) and processed in a Corbett Rotor-Gene 6000 (Qiagen, Hilden, Germany) at individually optimal protocols (Table 2). A no template control (NTC) with DNase-free water instead of cDNA was applied for each run. qPCR reactions for each sample and gene were run in duplicate to minimize dispensation artefacts. The deviation of *C*_t_ of the technical replicates was <0.3. If it was higher, data were discarded, and the run was repeated. The PCR cycles were run using automatic fluorescence emission following each PCR cycle and the amplification specificity was checked after each run by melting curve analysis. The primer sequences and conditions for qPCR are shown in Table 2; the denaturation temperature was always 95 °C and the extension was performed at 60 °C.

The primers were designed with the Primer BLAST tool from the National Center for Biotechnology Information (NCBI, Bethesda, MD, USA) according to known sequences from the basic local alignment search tool (BLAST) in the gene bank database of the NCBI and synthesized by Eurofins MWG (Ebersberg, Germany). The amplicons were sequenced again and the product sequences were verified by BLAST.

The quantification cycle and amplification efficiency of each amplification curve were determined using the Rotor Gene 6000 Series Software 1.7 (Corbett/Qiagen, Hilden, Germany). For analysis of the data, the ‘relative expression software tool’ (REST 2009-RG Mode, Qiagen, Hilden, Germany) established by Pfaffl et al. [42] was used to calculate the relative mRNA expression with reference to the control group incubated at 21% O_2_ for the CaCo-2 cells and to native tissue that was snap-frozen immediately after the isolation (i.e., not incubated in the Ussing chamber) for the lagomorph jejunum epithelium, respectively, and whose expression was set to 1. The *C*_t_ values set by the software were applied after checking them optically.

Normalization of the samples was achieved using the same amounts of RNA for processing and by normalizing the data for the target genes with the aid of the reference genes hypoxanthine guanine phosphoribosyltransferase 1 (HPRT1), peptidylprolyl-isomerase (PPI) A and B and β-actin. The reference genes have been proven to be stable under the experimental conditions applied in our study. Their stability was tested using the programs BestKeeper© (Version 1 by M.W. Pfaffl, Institute of Physiology, Center of Life and Food Sciences, TUM-Weihenstephan, Germany, 2004) and geNorm [43].

### 4.5. Statistics

The results are described as arithmetic means ± standard deviation (SD) or as box plot representing the median ± 10th, 25th, 75th, and 90th percentile. The significance is expressed as the probability of error (*p*). Regarding the Ussing chamber experiments, *N* indicates the number of animals used, and *n* represents the number of single epithelia for each treatment. For cell culture, each independent experiment with different stocks and passages of the cells is considered as a biological replicate (“*N*”), while technical replicates are considered as “*n*”. The data were pooled for each *N* for statistical analysis. The differences between the mean values of two groups were tested using the repeated-measures one-way analysis of variance (ANOVA) with a subsequent Holm–Sidak or Tukey test as appropriate (Sigma Plot 13.0, Systat Software, Erkrath, Germany). Gene expression levels were compared using a pair wise fixed reallocation randomization test using the REST software [42]. The differences were assumed to be statistically significant if *p* < 0.05.

## Figures and Tables

**Figure 1 ijms-20-04993-f001:**
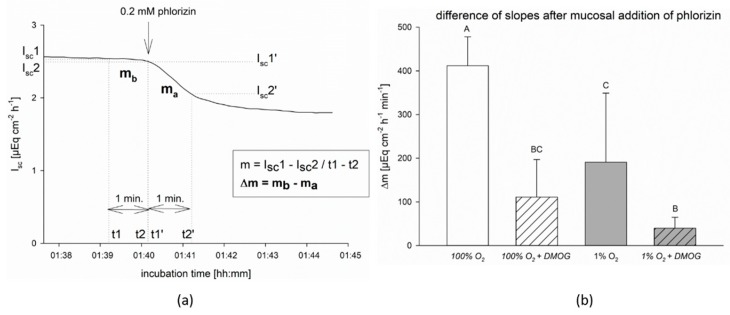
Activity of sodium-coupled glucose transporter SGLT1 in isolated lagomorph jejunum epithelia under hypoxia and preincubation with DMOG. After equilibration in the buffer solution, half of the epithelia were preincubated with 2 mM DMOG for 15 min. Subsequently, part of the control and part of the DMOG group was subjected to ‘hypoxia’. After 45 min of ‘hypoxia’, 0.2 mM phlorizin was added to the mucosal buffer solution and the slopes of I_sc_ for one minute before (m_b_) and after (m_a_) the addition were calculated as outlined in the Materials and Methods section and exemplified on the left hand (**a**). The difference in the slopes (Δm = m_b_ − m_a_) after mucosal addition of phlorizin (i.e., the effect of SGLT1 inhibition) was calculated and compared between the treatment groups. (**b**) shows that Δm was decreased by ‘hypoxia’ (grey bars) compared to the control group gassed with 100% oxygen (white bars) and also by preincubation with DMOG (hatched bars). Bars represent mean ± SD; one-way repeated measurements ANOVA with a subsequent Holm–Sidak test based on *N* = 6 (animals) (*n* = 10 (epithelia)), *p* < 0.05, different letters indicate significant differences between groups.

**Figure 2 ijms-20-04993-f002:**
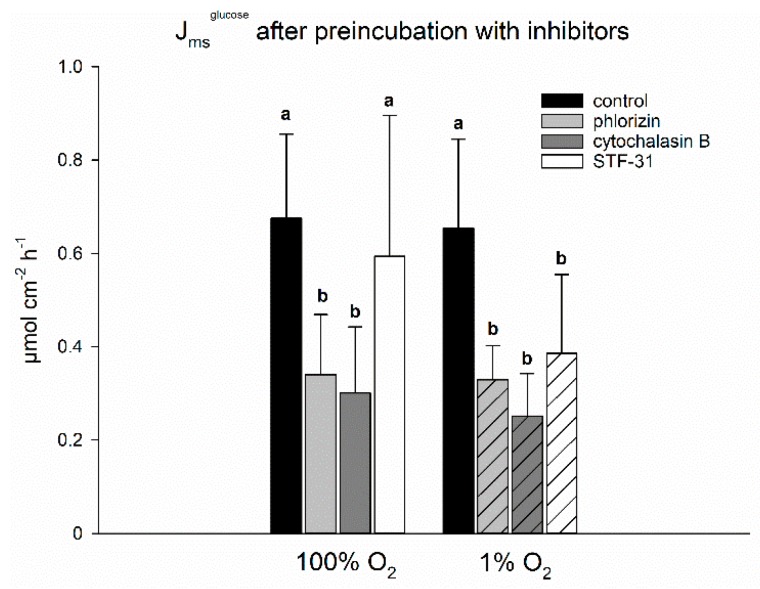
J_ms_^glucose^ across isolated lagomorph jejunum epithelia after the incubation with inhibitors for SGLT1 and glucose transporters GLUT2 and GLUT1. J_ms_^glucose^ was similar under hypoxia and gassing with 100% O_2_. While J_ms_^glucose^ was decreased significantly by phlorizin (inhibiting SGLT1, light grey bars) and cytochalasin B (inhibiting GLUT2, dark grey bars) under both gassing regimes, STF-31 (inhibiting GLUT1, white bars) had an effect under hypoxia (hatched bars) only. Bars represent mean ± SD; one-way repeated measurements ANOVA with a subsequent Holm–Sidak test based on *N* = 6 (*n* = 12), *p* < 0.05, different letters indicate significantly different flux rates within each group.

**Figure 3 ijms-20-04993-f003:**
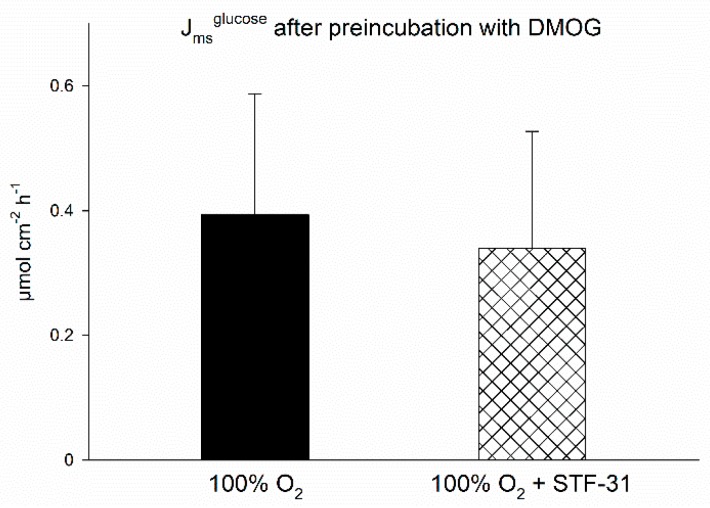
J_ms_^glucose^ across isolated lagomorph jejunum epithelia after preincubation with DMOG and with or without the GLUT1-inhibitor STF-31. Epithelia were incubated as described above but preincubated with 2 mM DMOG instead of simulating hypoxia. J_ms_^glucose^ was not significantly different between epithelia preincubated with DMOG only and epithelia incubated with DMOG and STF-31 (inhibiting GLUT1). There is no significant difference between the groups. Bars represent mean ± SD; *N* = 6 (*n* = 12) for DMOG only and *N* = 4 (*n* = 8) for DMOG + STF-31.

**Figure 4 ijms-20-04993-f004:**
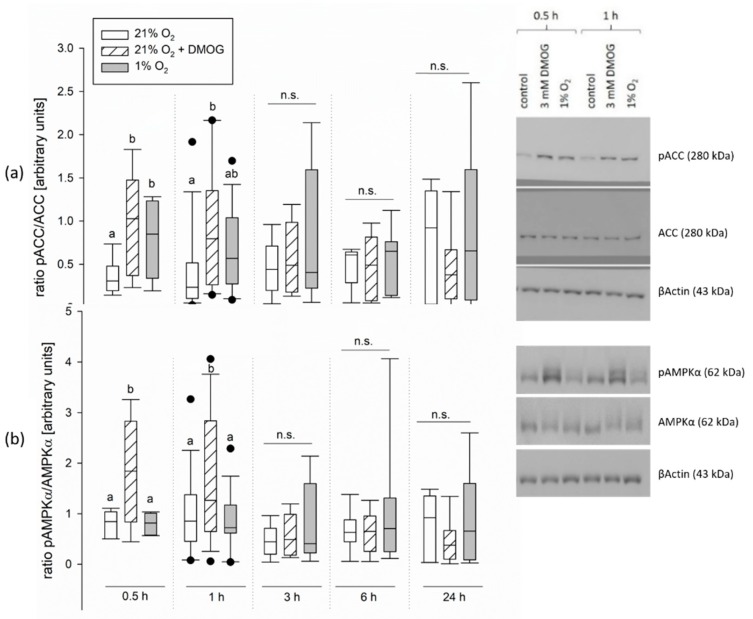
Activation of AMP-activated protein kinase (AMPK) in CaCo-2 cells as indicated by the phosphorylation of AMPKα and its target ACC. Cells were incubated under 21% O_2_ (white bars), with 3 mM DMOG (hatched bars) or kept at 1% O_2_ (grey bars) for 0.5, 1, 3, 6, or 24 h, as indicated at the x-axis. Total protein was extracted, separated by SDS-PAGE and AMPKα, ACC, pAMPKα, and pACC were detected using Western blot. (**a**) The ratio of pACC/ACC was increased after 0.5 h in both treatment groups and after 1 h of incubation with DMOG. Longer incubation times did not increase pACC compared to the control group incubated at 21% O_2_. (**b**) The ratio pAMPKα/AMPKα was increased after incubation with DMOG for 0.5 and 1 h as well. However, there was no significant increase after hypoxic incubation compared to the control group incubated at 21% O_2_. Longer incubation times had no effect. Boxes show the median, 10th, 25th, 75th, and 90th percentile plus error bars. Outliers are represented by dots. One-way repeated measurements ANOVA, *p* < 0.01, *N* = 4 (*n* = 8) for 6 h, *N* = 3 (*n* = 6) for 24 and 3 h, *N* = 5 (*n* = 10) for 1 and 0.5 h. Right side: representative blots after 0.5 and 1 h incubation time.

**Figure 5 ijms-20-04993-f005:**
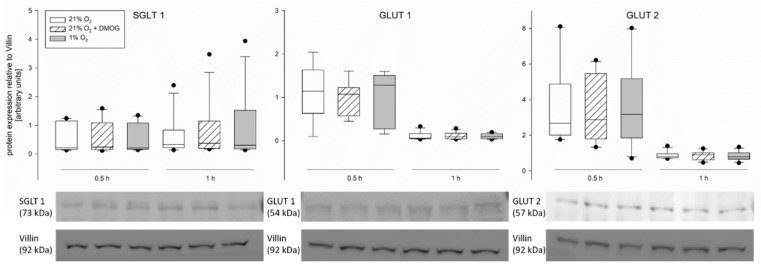
Protein expression of SGLT1, GLUT1, and GLUT2 in CaCo-2 cells detected by Western blot. Cells were incubated at 21% O_2_ (white bars) or at 21% O_2_ with 3 mM DMOG (hatched bars) or subjected to hypoxia (grey bars). There is no significant difference between the treatments. Representative blots are shown below. Boxes show the median, 10th, 25th, 75th, and 90th percentile plus error bars. Outliers are represented by dots. Paired t-test, *N* = 5 (*n* = 10) for 0.5 h and *N* = 6 (*n* = 12) for 1 h.

**Figure 6 ijms-20-04993-f006:**
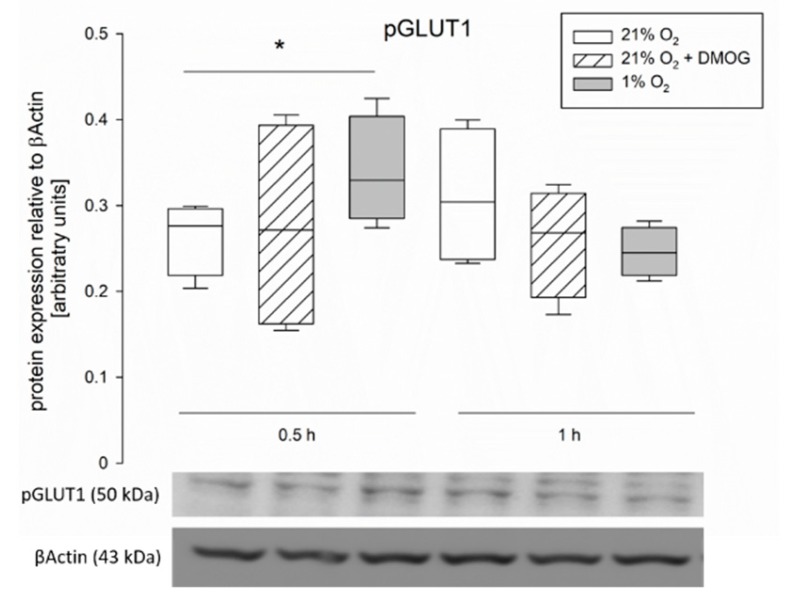
Protein expression of pGLUT1 in CaCo-2 cells detected by Western blot. Cells were incubated at 21% O_2_ (white bars) or at 21% O_2_ with 3 mM DMOG (hatched bars) or subjected to hypoxia (grey bars). There is a significant difference after 0.5 h of incubation at 1% O_2_ compared to 21% O_2_. Representative blots are shown below. Boxes show the median, 10th, 25th, 75th, and 90th percentile plus error bars. Outliers are represented by dots. Paired t-test, *p* < 0.05, *N* = 4.

**Figure 7 ijms-20-04993-f007:**
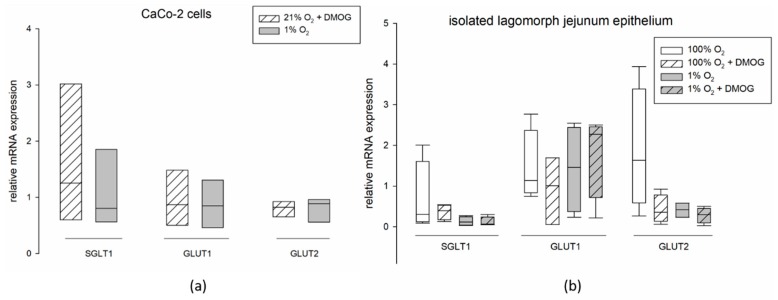
mRNA expression of SGLT1, GLUT1, and GLUT2 in CaCo-2 cells and isolated lagomorph jejunum epithelium. (**a**) Cells were incubated with 3 mM DMOG at 21% O_2_ (hatched bars) or under 1% O_2_ (grey bars) for 1 h. The mRNA expression was measured relative to the expression levels in the control group (21% O_2_), whose expression was set to 1. (**b**) Isolated lagomorph jejunum epithelia were incubated with or without 2 mM DMOG under 100% O_2_ or 1% O_2_ gassing for 1 h. The mRNA expression was measured relative to the expression in native tissue (i.e., not incubated in the Ussing chamber), whose expression was set to 1. There is no significant difference between the different incubation conditions. Boxes show the median, 10th, 25th, 75th, and 90th percentile plus error bars. Paired t-test, *N* = 3 for CaCo-2 cells and *N* = 4 for isolated epithelia.

**Table 1 ijms-20-04993-t001:** Antibodies used for Western blot.

Primary Antibody	Manufacturer	Dilution	Secondary Antibody	Manufacturer	Dilution
Acetyl-CoA Carboxylase (C83B10) Rabbit mAb 3676	Cell Signaling Technology Europe B.V., Frankfurt/Main, Germany	1:1000	Anti-rabbit IgG, HRP-linked Antibody 7074	Cell Signaling Technology Europe B.V., Frankfurt/Main, Germany	1:5000
Phospho-Acetyl-CoA Carboxylase (Ser79) (D7D11) Rabbit mAb 11818		1:1000			
AMPKα (D5A2) Rabbit mAb 5831		1:1000			
Phospho-AMPKα (Thr172) (40H9) Rabbit mAb 2535		1:1000			
SGLT1, rabbit polyclonal Antibody, ABIN364451	Antibodies online, Aachen, Germany	1:500	Anti-rabbit IgG, HRP-linked, sc-2077	Santa Cruz Biotechnology, Heidelberg, Germany	1:5000
GLUT1, sheep polyclonal antibody, ab54263	Abcam, Berlin, Germany	1:1000	Anti-sheep IgG, HRP-linked, preadsorbed, ab195176	Abcam, Berlin, Germany	1:10000
pGLUT1 (Ser226), rabbit polyclonal antibody, ABN991	Merck Millipore, Darmstadt, Germany	1:500	Anti-rabbit IgG, HRP-linked, sc-2077	Santa Cruz Biotechnology, Heidelberg, Germany	1:5000
GLUT2 rabbit polyclonal antibody, PA5-77459	ThermoFisher Scientific, Dreieich, Germany	1:200	Anti-rabbit IgG, HRP-linked, sc-2077	Santa Cruz Biotechnology, Heidelberg, Germany	1:5000
β-Actin, mouse monoclonal antibody, sc-47778	Santa Cruz Biotechnology, Heidelberg, Germany	1:1000	Anti-mouse IgG, HRP-linked, A16072	Invitrogen, Dreieich, Germany	1:5000
Villin, rabbit polyclonal antibody, PA5-78222	ThermoFisher Scientific, Dreieich, Germany	1:1000	Anti-rabbit IgG, HRP-linked, sc-2077	Santa Cruz Biotechnology, Heidelberg, Germany	1:5000

**Table 2 ijms-20-04993-t002:** Primers used for RT-qPCR.

	Gene Name	Gene Bank Accession Number	Primer Sequence (5′–3′)	Annealing Temperature (°C)	Amplicon Length (bp)
CaCo-2	HPRT1	NM_000194.3	F: ATGGACAGGACTGAACGTCTT R: TGTAATCCAGCAGGTCAGCA	57	118
	PPIA	NM_021130.5	F: GCCGAGGAAAACCGTGTACT R: CTGCAAACAGCTCAAAGGAGAC	59	106
	GLUT1	NM_006516.3	F: GAACTCTTCAGCCAGGGTCC R: ACCACACAGTTGCTCCACAT	60	114
	GLUT2	NM_000340.2	F: CAATGCACCTCAACAGGTAATAA R: AGATTGTGGGCAGTTCATCTGT	57	119
	SGLT1	NM_000343.3	F: AAGACCACCGCGGTCAC R: AAACATAGCCCACAGTCCGA	57	120
Oryctolagus cuniculus	PPIB	DQ237914.1	F: CGGGTGGTCTTTGGTCTCTT R: TGTAGCCAAATCCTTTCTCCCC	60	90
	β-actin	NM_001101683.1	F: AAACTGGAACGGTGAAGGTGA R: ACAATCAAAGTCCTCGGCCA	58	89
	HPRT1	NM_001105671.1	F: AGACCTTGCTTTCCTTGGTCA R: TCCAACAAAGTCTGGCCTGTA	58	106
	GLUT1	NM_001105687.1	F: GTATGTGGAGCAACTGTGCG R: TCCGGCCTTTAGTCTCAGGA	59	104
	GLUT2	XM_017346915.1	F: TCACTGCTCTCTCCGTATTCC R: ATCCAGTAGCACGAGTCCCA	59	118
	SGLT1	NM_001101692.1	F: GAGAGTCAACGAGCCTGGAG R: CCCGGTTCCATAGGCAAACT	60	93

F: forward primer; R: reverse primer; bp: base pairs.

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
