# Peer review of "The Fast Lane of Hypoxic Adaptation: Glucose Transport Is Modulated via A HIF-Hydroxylase-AMPK-Axis in Jejunum Epithelium"

_ijms, 2019, doi:10.3390/ijms20204993_

Round 1
Reviewer 1 Report
This study by Dengler and Gabel examines the role of the AMP- activated protein kinase (AMPK) in regulating glucose transport over the jejunum epithelium in conditions of low oxygen. This study follows from the group’s previous work published in the Journal of Applied Physiology in 2017 where they have used the same experimental system to demonstrate that the hypoxia-dependent decrease in glucose transport across the jejunum epithelium is dependent on AMPK. The current study expands on these data demonstrating that the pan-hydroxylase inhibitor, DMOG, broadly phenocopies the results observed with physiological hypoxia, indicating that hypoxia dependent AMPK activation in the jejunum epithelium is dependent on inhibition of oxygen-sensitive PHD enzymes.
Specific Comments
In the jejunum epithelium experiments using the Ussing chamber why was 100% O2 used as the control? There are studies demonstrating that the signalling pathways being studied are sensitive to hyperoxia – please comment. Does the observation that DMOG and hypoxia have an additive effect in Figure 1B not actually indicate that there are PHD-independent signalling pathways indiced by hypoxia that can contribute to the phenotype. In the AMPK activation assays (figure 4) why is ACC used, rather than directly measuring activated AMPK as described in the Journal of Applied Physiology 2017 article from the same group. In Figure 5 it would be beneficial to include representative western blots like in Figure 4.
Author Response
This study by Dengler and Gabel examines the role of the AMP- activated protein kinase (AMPK) in regulating glucose transport over the jejunum epithelium in conditions of low oxygen. This study follows from the group’s previous work published in the Journal of Applied Physiology in 2017 where they have used the same experimental system to demonstrate that the hypoxia-dependent decrease in glucose transport across the jejunum epithelium is dependent on AMPK. The current study expands on these data demonstrating that the pan-hydroxylase inhibitor, DMOG, broadly phenocopies the results observed with physiological hypoxia, indicating that hypoxia dependent AMPK activation in the jejunum epithelium is dependent on inhibition of oxygen-sensitive PHD enzymes.
Dear reviewer,
thank you very much for your time and effort reading our manuscript and your suggestions to improve it. We implemented the changes asked for and tried to answer your questions in the following.
Specific Comments
In the jejunum epithelium experiments using the Ussing chamber why was 100% O2 used as the control? There are studies demonstrating that the signalling pathways being studied are sensitive to hyperoxia – please comment.
Answer:
In Ussing chamber experiments, gassing with 100% oxygen or carbogen (95% oxygen) is commonly used. This is due to practical reasons:
The oxygen has to be dissolved physically in the buffer solution and cannot be transported as effectively as in the blood; therefore, it is assumed that the actual amount of oxygen near and inside the tissue is less than 100%. It has to be borne in mind that what is commonly considered as ‘isolated epithelium’ still consists of several layers (basal membrane, epithelium, mucus, connective tissue…) in contrast to cultured cells growing in monolayer. The only possibility for oxygen reaching the cells is via diffusion of the physically dissolved oxygen (see above).
Therefore, only a small part of the 100% oxygen used for gassing will reach the epithelial cells.
We agree with you that hyperoxia should be considered more intensively regarding experimental conditions. However, this is also true for cell culture, where the commonly used 21% oxygen are probably hyperoxic as well. Testing an additional “normoxic” group is beyond the scope of the current manuscript but your comment further encourages us to do so in future experiments.
Although we cannot determine the actual oxygen levels in the epithelium, the concentrations used in our experiments clearly induce functional changes, therefore we consider the difference in oxygen supply to be biologically significant, which was the main aim of the current study.
Does the observation that DMOG and hypoxia have an additive effect in Figure 1B not actually indicate that there are PHD-independent signalling pathways indiced by hypoxia that can contribute to the phenotype.
Answer:
Thank you for this comment. Reviewer 2 also commented on the statistics described for these experiments, thus we changed the statistical test used. In a one-way repeated measurements ANOVA with subsequent Holm-Sidak pairwise comparison, we found significant differences between the group incubated under hypoxia but without DMOG and the hypoxic group incubated with DMOG, but no difference between the hypoxic group without DMOG and the group incubated with 100% oxygen and DMOG. Therefore, we assume the effects of hypoxia and DMOG to be similar. In the epithelia incubated under hypoxia and with DMOG these (similar) effects might be enhanced by both stimuli affecting the same pathways simultaneously. If there was an additional (PHD-independent) effect of hypoxia, we would have expected the effect of hypoxia alone to be different from that of DMOG alone.
In the AMPK activation assays (figure 4) why is ACC used, rather than directly measuring activated AMPK as described in the Journal of Applied Physiology 2017 article from the same group.
Answer:
Thank you very much for your comment. We find the phosphorylation of the AMPK target protein ACC more meaningful, as it indicates not only phosphorylation, but also biological activity of AMPK. The phosphorylation grade of ACC is widely used to assess AMPK activation (e.g. Faubert et al. 2013, Mungai et al. 2011). Thus, we consider this to be rather an improvement of our methodology compared to our earlier publication. However, we also measured pAMPKα and agree with you that the phosphorylation grade of AMPK itself is also of interest, especially as the results are quite interesting compared to ACC. We added this data to the manuscript (please refer to Fig. 4 and l. 181ff.)
In Figure 5 it would be beneficial to include representative western blots like in Figure 4.
Answer:
We added representative blots to Fig. 5 as you suggested.
We also corrected an error that was identified when preparing the representative blots. For the analysis of the glucose transport protein, we normalized the data using Villin instead of βActin, as this larger protein could be stained on the same blot without interfering with the transporter staining. We apologize for this mistake.
References
Faubert, B.; Boily, G.; Izreig, S.; Griss, T.; Samborska, B.; Dong, Z.; Dupuy, F.; Chambers, C.; Fuerth, B.J.; Viollet, B.; et al. AMPK is a negative regulator of the Warburg effect and suppresses tumor growth in vivo. Cell Metab. 2013, 17, 113–124, doi:10.1016/j.cmet.2012.12.001. Mungai, P.T.; Waypa, G.B.; Jairaman, A.; Prakriya, M.; Dokic, D.; Ball, M.K.; Schumacker, P.T. Hypoxia triggers AMPK activation through reactive oxygen species-mediated activation of calcium release-activated calcium channels. Mol. Cell. Biol. 2011, 31, 3531–3545, doi:10.1128/MCB.05124-11.
Reviewer 2 Report
The manuscript by Dengler et al. examines the adaptability of intestinal epithelial to potential events of hypoxia concluding that prolyl hydroxylases have a direct effect on the activation of AMPK which impart beneficial effects by promoting the transport of glucose across the epithelium. They argue that these short-term responses precede the long-term transcriptional responses provided by HIFs. Indeed, it is well established that inhibition of PHDs increases HIF-mediated enhancement of glycolysis.
The manuscript is mostly supportive of previous reports. For example, this paper argues that PHDs have a direct effects on the activation of AMPK and such data previously published by others (ie in cardiomyocytes JCMM 2012 16(9): 2049-59).
2) The author previously showed significant changes showing hypoxia pretreatment (1%, 45 min) significantly decreased the short circuit current effects of SGLT-1 inhibition (ref 5), also illustrated in Figure 1B. DMOG treatment itself also significantly decreases its effect. However, the authors mention a ‘larger decrease… by DMOG incubation under hypoxia… compared to hypoxia alone… (P3 Lines 120-122) there is no statistical reference to this additive effect if significant.
3) Interesting, by using specific inhibitors to glucose transporters, the authors demonstrate hypoxic differences only GLUT1 inhibition (figure 3) while PHDi (DMOG) had no effect. Again, it is hard to resolve the molecular/biochemical changes unique to DMOG which affecting the functional changes of the jejunum epithelium. Indeed, There is no direct proof that AMPK, or its downstream players are responsible for the observed decreased difference in glucose flux created by DMOG.
4)There is no indication that the ‘unique’ effects by DMOG are due to changes in the protein (Fig 5) or mRNA (Fig. 6) expression of glucose transporters and no additional data to determine the mechanism unique to DMOG-AMPK affects dampening the importance of this manuscript.
To examine the potential HIF-independent effects of DMOG, use of PHD knockdowns/outs, HIF knokdowns/outs (siRNAs, CRISPERS), and in contrast to expressing stable-HIFs (independent of Hypoxia then) in their CaCo-2 cell model would validate the observations and permit additional molecular studies.
5) Well-labelled and formatted figures with representative Western blots would make this an easier to follow and stronger manuscript.
6) For normal distribution data and variation, we expect to report SD (measure of variability) measuring the sampling and helps compare differences between different treatments which also had different sizes. SEM deals with population.
Minor:
1) The baseline responses to any of Ussing chamber experiments is the classically gasses with 100% oxygen. It is well recognized that hyperoxia also alters cellular and tissue responses, for example by inducing HIF-alpha protein levels. Have they tried gassing the jejunum with physiological levels of oxygen?
2) Figure legends are hard to follow. Providing titles and labeling each part within each figure may be helpful.
Author Response
The manuscript by Dengler et al. examines the adaptability of intestinal epithelial to potential events of hypoxia concluding that prolyl hydroxylases have a direct effect on the activation of AMPK which impart beneficial effects by promoting the transport of glucose across the epithelium. They argue that these short-term responses precede the long-term transcriptional responses provided by HIFs. Indeed, it is well established that inhibition of PHDs increases HIF-mediated enhancement of glycolysis.
The manuscript is mostly supportive of previous reports. For example, this paper argues that PHDs have a direct effects on the activation of AMPK and such data previously published by others (ie in cardiomyocytes JCMM 2012 16(9): 2049-59).
Dear reviewer,
thank you very much for your time and effort reading our manuscript. We tried to implement all your comments and answered your questions in the following. We hope we could clarify all points satisfyingly and you’ll find the manuscript improved now.
2) The author previously showed significant changes showing hypoxia pretreatment (1%, 45 min) significantly decreased the short circuit current effects of SGLT-1 inhibition (ref 5), also illustrated in Figure 1B. DMOG treatment itself also significantly decreases its effect. However, the authors mention a ‘larger decrease… by DMOG incubation under hypoxia… compared to hypoxia alone… (P3 Lines 120-122) there is no statistical reference to this additive effect if significant.
Answer:
Thank you for this comment. From our previous way of presenting the data and statistics in the manuscript it was actually not clear that there is also a statistical difference between these groups. Your comment made us reconsider the statistics and we decided to use a one-way repeated measurements ANOVA instead. This makes it easier to see the significant differences in the figure. Please refer to l. 116ff. and Fig. 1b.
3) Interesting, by using specific inhibitors to glucose transporters, the authors demonstrate hypoxic differences only GLUT1 inhibition (figure 3) while PHDi (DMOG) had no effect. Again, it is hard to resolve the molecular/biochemical changes unique to DMOG which affecting the functional changes of the jejunum epithelium. Indeed, There is no direct proof that AMPK, or its downstream players are responsible for the observed decreased difference in glucose flux created by DMOG.
Answer:
You are right, we cannot deduce an involvement of AMPK from the flux experiments and we do not postulate this in the discussion. This would have been legitimate only if we had been successful reproducing the inhibitory effect of STF-31 after DMOG incubation.
However, we found differences in the flux rates after DMOG incubation in general, which we cannot definitely assign to an effect of DMOG or individual characteristics of the animals used for this experimental series.
The missing effect of DMOG might be a hint that PHDs / AMPK are not involved in this recruitment, but it is also possible that there are other, stronger effects of DMOG on transepithelial glucose transport, as indicated in l. 158ff.
However, we still consider these experiments to be of interest to the scientific community, as they indicate a recruitment of GLUT1 that has not been shown before to our knowledge.
We also added our newest results showing an increased phosphorylation of GLUT1 under hypoxia but not DMOG incubation compared to the control group (see Fig. 6), which also indicate that this recruitment of GLUT1 is mediated hydroxylase-independently under hypoxia. This new aspect is discussed in l. 293ff.
4)There is no indication that the ‘unique’ effects by DMOG are due to changes in the protein (Fig 5) or mRNA (Fig. 6) expression of glucose transporters and no additional data to determine the mechanism unique to DMOG-AMPK affects dampening the importance of this manuscript.
To examine the potential HIF-independent effects of DMOG, use of PHD knockdowns/outs, HIF knokdowns/outs (siRNAs, CRISPERS), and in contrast to expressing stable-HIFs (independent of Hypoxia then) in their CaCo-2 cell model would validate the observations and permit additional molecular studies.
Answer:
Thank you for your suggestions. We will keep them in mind for future experiments. Regarding the missing changes in protein and mRNA expression, this fits quite well with the mode of action of AMPK, i.e. phosphorylation of the proteins but no changes in their expression. We pointed out this possibility in the discussion (see l. 276ff.) and also indicated the need for further studies using knock outs or siRNA (see l.333f.).
Additionally, we also extended our efforts to detect the phosphorylated proteins and succeeded in staining pGLUT1 which was actually increased after 30 minutes of hypoxia in CaCo-2 cells but not after DMOG treatment. This could be a first indicator for additional pathways influencing the epithelial adaptation to hypoxia. Please refer to l. 215ff. and l. 287ff. for the description and the discussion of this new additional result.
5) Well-labelled and formatted figures with representative Western blots would make this an easier to follow and stronger manuscript.
Answer:
Thank you for this comment. We added pictures of representative blots to Fig. 5.
We also corrected an error that was identified when adding the representative blot. For the analysis of the glucose transport protein, we normalized the data using Villin instead of βActin, as this larger protein could be stained on the same blot without interfering with the transporter staining. We apologize for this mistake.
We tried to improve labelling of the figures as well.
6) For normal distribution data and variation, we expect to report SD (measure of variability) measuring the sampling and helps compare differences between different treatments which also had different sizes. SEM deals with population.
Answer:
Thank you for pointing this out to us. We replaced SEMs with SDs in the bar charts (see Fig. 1-3).
Minor:
1) The baseline responses to any of Ussing chamber experiments is the classically gasses with 100% oxygen. It is well recognized that hyperoxia also alters cellular and tissue responses, for example by inducing HIF-alpha protein levels. Have they tried gassing the jejunum with physiological levels of oxygen?
Answer:
In Ussing chamber experiments, gassing with 100% oxygen or carbogen (95% oxygen) is commonly used. This is due to practical reasons:
The oxygen has to be dissolved physically in the buffer solution and cannot be transported as effectively as in vivo by the blood; therefore, it is assumed that the actual amount of oxygen near and inside the tissue is less than 100%. It must be borne in mind that what is commonly considered as ‘isolated epithelium’ still consists of several layers (basal membrane, epithelium, mucus, connective tissue…) in contrast to cultured cells growing in monolayer. The only possibility for oxygen reaching the cells is via diffusion of the physically dissolved oxygen (see above).
Therefore, only a small part of the 100% oxygen used for gassing will reach the epithelial cells.
We agree with you that hyperoxia should be considered more intensively regarding experimental conditions. However, this is also true for cell culture, where the commonly used 21% oxygen are probably hyperoxic as well. Testing an additional “normoxic” group is beyond the scope of the current manuscript but your comment encourages us to do so in future experiments.
Although we cannot determine the actual oxygen levels in the epithelium, the concentrations used in our experiments clearly induce functional changes, therefore we consider the difference in oxygen supply to be biologically significant, which was the main aim of the current study.
2) Figure legends are hard to follow. Providing titles and labeling each part within each figure may be helpful.
Answer:
Thank you for the comment, we tried to improve labelling of the figures.
Round 2
Reviewer 2 Report
The author's have improved the present form of the manuscript by performing stronger statistical analyses and including Western blots in there figures.
They have also added phospho-of GLUT1 data which suggests additional 'immediate' changes in the regulation of glucose transport by hypoxia.
However, as this reviewer noted no additional experiments were included to validate the suggested PHD-AMPK axis in mediating glucose transport. If no molecular studies ['use of PHD knockdowns/outs, HIF knokdowns/outs (siRNAs, CRISPERS), and in contrast to expressing stable-HIFs (independent of Hypoxia)'], then confirmation using another PHD inhibitor (FG-4497) would validate the AMPK axis as proposed,
minor:
However, the Western blots are missing ladder marker sizes.
Figure 7: mRNA expression is still not statistically quantified; no p values are provided
Author Response
The author's have improved the present form of the manuscript by performing stronger statistical analyses and including Western blots in there figures.
They have also added phospho-of GLUT1 data which suggests additional 'immediate' changes in the regulation of glucose transport by hypoxia.
Dear reviewer,
thank you again for your comments on our manuscript. We are happy, that the changes we made satisfy you and you find the manuscript improved now. In the manuscript, you will find our latest changes in blue and those from the first revision still in red. In the following, we would like to respond to your remaining concerns.
However, as this reviewer noted no additional experiments were included to validate the suggested PHD-AMPK axis in mediating glucose transport. If no molecular studies ['use of PHD knockdowns/outs, HIF knokdowns/outs (siRNAs, CRISPERS), and in contrast to expressing stable-HIFs (independent of Hypoxia)'], then confirmation using another PHD inhibitor (FG-4497) would validate the AMPK axis as proposed,
As indicated in our previous response to this point, we consider the creation of several knockouts to be beyond the scope of this study, especially with respect to its functional focus rather than a sole mechanistic approach. We thank you very much for suggesting using an additional inhibitor which can be performed more easily. However, we regret to inform you that we had only a very short time period for the revision (10 days) which did not allow for additional experiments. We checked prolongation of revision time with the editorial office, but it was not possible to extend this period. Thus, we added this thought to the discussion and emphasized the limitations of our study (please refer to l. 310ff. and l. 333ff.). We sincerely hope that you will be able to agree to this compromise.
minor:
However, the Western blots are missing ladder marker sizes.
Thanks for pointing this out, we corrected it in the figures 4 - 6.
Figure 7: mRNA expression is still not statistically quantified; no p values are provided
We did not add a description of the statistics for the mRNA and total protein expression, because there was no statistically significant difference. Now we added the test we used (paired t-test) to the figure legends as well, please refer to l. 214 and l. 239f.
Yours sincerely
Franziska Dengler
Round 3
Reviewer 2 Report
The author's have addressed the concerns.
Author Response
Dear reviewer,
thank you very much for reviewing our work. We appreciate your time and effort and will try to implement your suggestions in future studies.
Best regards
Franziska Dengler